# Post-Traumatic Single-Digit Onychomycosis

**DOI:** 10.3390/jof9030313

**Published:** 2023-03-02

**Authors:** Eckart Haneke, Ganna Stovbyr

**Affiliations:** 1Department Dermatol Inselspital, University of Berne, 3010 Bern, Switzerland; 2Private Dermatol Practice Dermaticum, 79098 Freiburg, Germany; 3Centro Dermatol Epidermis, Instituto CUF, 4460-188 Matosinhos, Porto, Portugal; 4Euroderm Clinic, 02000 Kyiv, Ukraine

**Keywords:** onychomycosis, single digit, trauma, histopathology, diagnostic problems, therapeutic dilemma

## Abstract

Onychomycoses are a group of fungal nail infections commonly classified either according to the pathogenic fungus, to the duration of the disease or to the mode of fungal invasion. Most cases are diagnosed clinically, although there is a general consensus that the pathogen should be identified prior to initiating a treatment. However, this is often difficult as the classical mycologic methods of direct microscopy and culture frequently remain negative. We came across a particular subset of onychomycoses, which posed extreme diagnostic and therapeutic challenges. Over a period of 15 years, 44 patients were seen in specialized nail clinics with a single nail dystrophy that was examined and treated in vain by many practitioners and dermatologists prior to their consultation. Of the forty-four cases, thirty-nine patients had a fingernail affected and five had a toenail affected. The nail was almost completely onycholytic, the nail bed visibly keratotic, the proximal nail fold smooth and shiny and slightly swollen. All patients except five brought the results of negative mycologic cultures. Thirty-four patients had received antifungal therapy, mostly topical, as a single nail would not qualify for systemic treatment according to most national and international guidelines. The diagnosis was finally confirmed by histopathology of the nail plate showing an invasive onychomycosis in all cases. After nail avulsion and combined topical and systemic antifungal therapy, thirty-six patients were cured, three were lost from follow-up, and five showed improved nails but not a complete clinical and mycologic cure. A single-digit nail disease raises the suspicion of a tumor or a trauma; although, in rare cases, diseases normally affecting several nails may only affect a single nail. Such a case should prompt the clinician to ask for a previous trauma to this digit and to intensify the search for a specific pathogen. This study also underlines the importance of histopathology for the diagnosis of onychomycoses.

## 1. Introduction

Onychomycoses are chronic infections of the nail unit with fungi of various species that are usually divided into dermatophytes, yeasts and non-dermatophyte molds. The former are generally accepted as nail pathogens, whereas the many different yeasts and molds often found in pathologic nail specimens require rigorous criteria for being acknowledged as the cause of an onychomycosis. Independent of the species, the clinical changes may be very similar and rarely permit the pathogen or group of pathogens to be determined. Some nail diseases are clinical mimickers of onychomycoses, particularly nail psoriasis and the asymmetric gait nail unit syndrome (AGNUS) [1], but also nail lichen planus, some nail eczemas and dystrophic changes due to peripheral vascular impairment and old age. Most of these disorders affect several nails, but, very rarely, a single nail is altered, posing considerable diagnostic difficulties. Predisposing factors for fungal nail infections are genetic susceptibility to fungal infection, which is an autosomal dominant trait [2], nail trauma, peripheral vascular insufficiency, peripheral neuropathy, immunosuppression, sports activities and many more [3]. Onychomycoses are not only aesthetically embarrassing but also have a serious impact on the quality of life [4], and may pose a true medial risk, particularly in diabetics, patients under immunosuppression and cancer therapy [5]. Further, they may be the portal of entry for streptococcal cellulitis and the origin of recurrences for tinea pedum [6].

Although trauma in general is a well-known predisposing factor for the development of fungal nail infections, a literature search in PubMed did not yield any article on onychomycosis after direct trauma. Alterations of a single nail, particularly of a fingernail in a female or a toenail in a male patient, should prompt the clinician to specifically ask for a trauma to this digit and to add histopathology of the nail or another sensitive laboratory method to the routine direct microscopy and culture for the confirmation of the diagnosis of a fungal nail infection.

The exact mechanism, by which a trauma that might have been sustained years ago, that renders this nail particularly susceptible for fungal infection is not known. It might be hypothesized that post-traumatic neuro-vascular alterations may impair the blood supply and reduce the immune response to fungal infections in the affected region. In extreme cases, a simple trauma, such as a nail biopsy, may even lead to reflex sympathetic dystrophy, a variant of the complex regional pain syndrome [7].

The role of a single trauma to a digit for the development of an onychomycosis has, to the best of our knowledge, not systematically been investigated. This series of random observations underlines its importance. Larger studies are warranted.

## 2. Patients and Methods

Over a period of 15 years (2007–2021), 44 patients consulted us in specialized nail clinics in Berne, Freiburg, Ghent, Porto and Kyiv (in the Department of Dermatology, Inselspital, University of Berne, Switzerland, in a private dermatology clinic in Freiburg, Germany, in the Department of Dermatology, University Hospital Ghent, Gent, Belgium, in the large practice clinic of Centro de Dermatología Epidermis, Instituto CUF, Senhora da Hora, Matosinhos, Porto, Portugal, and 2 were seen in a large Dermatology Center in Kiyv, Ukraine). Their age ranged from 22 to 68 years. More than two thirds of the patients were women. All of them reported a history of multiple physician visits for the nail problem of a single digit, the vast majority of a finger, only 5 of a single toe. They all had undergone multiple clinical and laboratory examinations including blood tests, onychoscopy, direct microscopy after KOH clearing and mycologic culture (Figure 1). In 40 patients, no fungus had been proven by microscopy and culture, 2 had non-dermatophyte molds considered non-pathogenic, and 2 had grown *Trichophyton rubrum* (Table 1).

The inclusion criteria for this observational study were positive history of trauma to the affected digit and proof of fungal infection by mycological culture and/or histopathology. There were no specific exclusion criteria (Figure 1). All patients gave their verbal permission for the researchers to use the information and photographs (this is part of the admission procedure). 

The clinical findings were very similar in the fingernails. There was a very high degree of onycholysis, usually over 75% of the nail field, but the nail was often completely detached, with the onycholysis reaching under the proximal nail fold. In individuals with skin type 4 and higher, the distal portion of the proximal nail fold was hyperpigmented. The nail plate was left in about half of the patients, the others had either cut the nail short or it was destroyed by the disease process. The nail bed was moderately hyperkeratotic and often shorter than normal. The proximal nail fold was swollen and soft. Its skin was smooth and shiny; however, the cuticle was intact in most patients (Figure 2).

The single toenails did not show characteristic alterations except that there was no accompanying tinea pedis. Swelling of the proximal nail fold was not present or very light (Figure 3).

Despite the patients’ claim that a mycosis had been ruled out several times, the onycholytic nail was clipped off and submitted for histopathologic examination with PAS stain. Scrapings from the keratotic nail bed were sent for direct microscopy and mycologic culture. Two patients had a punch biopsy performed and the slides were stained with Grocott’s silver methenamine method.

## 3. Results

The histopathology of all nail specimens showed invasive fungal filaments. Many of them were slender septate hyphae (Figure 4), but 18 of them showed filaments of varying diameters with short segments, often ampullar dilatations, and some had very large spore-like elements (Figure 5). These fungal structures were also identified in H&E stains in 14 of these cases (Figure 6). The punch biopsy specimens demonstrated an acanthotic and hyperkeratotic nail bed with invasive slender hyphae (Figure 7). Direct microscopy of the KOH preparations was positive in 27 cases and mainly showed filaments interpreted as fungal elements. Culture was positive in 30 cases with 15 growing dermatophytes (14 *T rubrum*, 1 *T mentagrophytes*) and 15 with a non-dermatophyte mold, mainly *Fusarium solani* and *oxysporum* complex. In a retrospective analysis of the nail photographs, no clinical difference was seen between the dermatophyte and *Fusarium* infections except for two cases with a white superficial onychomycosis of a big toe (Figure 8).

Thirty-nine patients agreed to nail avulsion plus combined treatment with topical ciclopirox in chitosan nail varnish (Ciclopoli^®^) for a minimum of 9 months, as well as 250 mg of terbinafine daily for a period of 4–6 months. The regrowing nail was healthy in 34 digits and had some distal involvement in another two. No clinical cure could be achieved in five digits; there was no clinical explanation such as particular comorbidities, medications, other risk factors or repeated trauma impairing cure. Three patients were lost from the follow-up.

## 4. Discussion

Our observations underlined four common problems seen in clinical practice:Difficulties in making the correct clinical diagnosis;Problems with confirming the diagnosis by laboratory methods;Efficacious treatment of onychomycoses;Particular therapeutic challenges of post-traumatic single-digit onychomycoses.

Onychomycoses are frequent and the most difficult to treat of all cutaneous mycoses [6,8]. They are known for their notorious chronicity and high relapse rate. They have a serious impact on the quality of life, as was evidenced in our patients, who consulted several specialists for their single nail, even though some of them expressed their frustration that their disease had been dismissed as a minor ailment because ‘only one nail’ was affected. Although often diagnosed on clinical grounds alone, the diagnosis of onychomycoses may not always be obvious and such cases may be a missed nail psoriasis or onycholysis due to AGNUS, and even ungual melanomas have been erroneously treated as an onychomycosis [8,9]. Virtually all national and international guidelines, therefore, recommend laboratory proof of a fungal infection before embarking on a systemic antifungal drug therapy. However, it is known that many cultures remain biologically false negative, which may have various reasons: the person taking the material for KOH and culture may be too timid to go to the very proximal portion of the infection for fear of hurting the patient, and therefore, the material may not contain viable fungi; the patient may have been treated with an antifungal drug that lasts long in the keratin of the nail plate and bed and prevents growth on culture plates, or the patient may use an OTC preparation with some antimycotic action; the culture medium may not be the correct one or contain a substance suppressing fungal growth; reading may be too early for a slow growing dermatophyte; a fast growing mold may have overgrown the real pathogen and the culture plate may have been discarded too early; and many more [10,11]. However, even the most reputed mycology labs have a considerable percentage of false negatives and histopathology has proven to be double as often positive as culture [12,13,14]. Many mycologists insist on culture as they consider it the only means to identify a fungus to the species level, even though a differentiation between dermatophytes, non-dermatophyte molds and yeasts is possible by histopathology of the nail in most cases [15,16,17]. Because of the high percentage of false negatives, many labs declare each fungus that grows on their plates as a (potential) nail pathogen, even though it is known that many fungi are simply colonizing the dead space under the onycholytic nail. Histopathology, in contrast, can show whether the fungus invades the subungual keratin and nail plate and is, thus, able to differentiate between true infection and colonization [14,15,16,17] (Table 2). It was shown in this observation to be the ultimate method to make the correct diagnosis.

Polymerase chain reaction (PCR) of material from diseased nails is more often positive than culture; however, it is not able to distinguish between fungal contamination and colonization. It also has the advantage of giving a result within a day compared to several weeks for the culture. It is, however, not available everywhere and expensive, although the price for the machinery has decreased. Matrix-assisted laser desorption/ionization–time of flight mass spectrometry (MALDI-ToF) is another modern technique using the specific spectrum of fungal proteins for identification; it requires a protein library of all potentially pathogenic fungi for comparison and enough fungal protein. Thus, it may be used directly for dermatophytoma and other extremely fungus-rich nail pieces, or a prior culture must be performed and the cultured fungus would then be rapidly identified by MALDI-ToF spectrometry. In situ hybridization would be the ideal method to both identify the fungus on a species level and to prove its localization in the nail; some research has been carried out [17], but this method is not yet available for routine diagnosis. Other sophisticated methods such as immunohistochemistry, immune adsorption, immunochromatography (approved in Japan) and ultraviolet excitation are not yet available for routine diagnosis [18,19] (Table 3).

Trauma has long been known to be a very important predisposing factor. It is, thus, crucial to ask the patient specifically for a sustained trauma, which, however, is not always remembered. The late Viennese mycologist Otto Male reported on a peculiar case at the Nidaros Dermatological Society, Trondheim, Norway, in the late 1960s: “A middle-aged man had a single-digit onychomycosis on his left 3^rd^ fingernail. He was treated with griseofulvin (the only available dermatophyte-active systemic antifungal drug at this time) for more than 2 years in vain. He then got his nail avulsed and was re-treated with griseofulvin again without success. Finally, the patient asked for an X-ray of the finger, which was first rejected because the mycologist knew that a nail mycosis can neither be diagnosed nor treated with a radiograph. As the patient insisted, the X-ray was done and revealed a tiny metal splinter in the proximal phalanx of this finger far away from the nail. It was removed and the onychomycosis healed rapidly by itself.” What sounds like a fairy tale may be seen as the basis of the observations described in this study. Virtually all patients had sustained an injury to the digit that they could remember, although sometimes only after repeated questioning. The most common trauma was a car door or a drawer crash injury, the latter particularly in women. Two female patients blamed aggressive manicure as the precipitating trauma. Four of the five single-toe onychomycoses were seen in men, and all had bumped their toe during barefoot leisure soccer playing. Toenail mycoses were usually associated with skin infection of the surrounding glabrous skin, often only seen as very discrete desquamation of the toe or sole of the foot. This was ruled out in all these patients.

Nail avulsion was the most severe iatrogenic trauma and should, thus, be an exception when dealing with an onychomycosis. However, there were cases where conservative treatments were doomed to fail because of the specific anatomical situation created by the infection. Nail avulsions render the nail even more susceptible to fungal infection and this has to be kept in mind in all cases where this procedure is planned. In our cases, all conservative measures had been in vain and it was, therefore, seen as indispensable. The method used was the proximal avulsion approach, which is the least traumatizing technique [22]. It has to be stressed that even in single nail onychomycosis, the combination of nail avulsion with topical treatment yields only poor results [23] and a triple combination with additional systemic therapy is required for a good result.

The clinical features of single fingernail mycosis are fairly characteristic: The nail plate is almost completely onycholytic, often considerably elevated from the nailbed. It is yellow and opaque. Probing often allows to go far proximal under the nail and proximal nail fold. This is often swollen, and its skin is shiny; however, in contrast to most other types of paronychia [24], the cuticle remains intact as the free margin of the nail fold is usually not thickened and rounded up.

The differential diagnosis of single-digit onychomycoses may be challenging if one relies on the result of mycologic examinations only. Especially in toenail involvement, the asymmetric gait nail unit syndrome (AGNUS) is an important mimicker. It shows onycholysis of variable degree and is usually mycologically negative but may sometimes be colonized with both pathogenic fungi as well as mere colonizers [1]. In these cases, orthopedic foot and sometimes leg and even vertebral column abnormalities have to be looked for. AGNUS is a very common condition more often seen in women than in men, as foot disease is more prevalent in females. It is even claimed that this nail disease is at least as frequent as toenail mycoses [1,25].

Another important differential diagnosis is single-digit nail psoriasis. It is rare and can be impossible to be ruled out on clinical grounds alone, as a psoriatic nail is prone to fungal infection [26]. We saw a patient with nail psoriasis who had one fingernail with psoriasis, one with onychomycosis and one with psoriasis plus onychomycosis, proven by histopathology. Another nail psoriasis patient was treated successfully for her ungual psoriasis, but one nail did not improve. Histopathology showed classical non-dermatophyte onychomycosis with many intraungual Munroe abscesses and invasive fungi both around and in these areas.

One striking feature of the single-digit post-traumatic onychomycosis was its recalcitrant nature. Most patients reported that they had had multiple diagnostic tests and despite negative mycology, at least one course of antifungal treatment. This was usually topical, but several patients had also received systemic or combined topical and systemic antifungal therapy. Considering the clinical features of this particular type of onychomycosis, its treatment-refractive nature was not surprising. The nail was almost completely detached from the nail bed and most of the matrix. Topical treatment was, therefore, hampered by inaccessibility of the infected nail bed and matrix under the onycholytic nail and proximal nail fold by the drug, whether it was a varnish, a solution or an ointment. Systemic drugs cannot reach into the nail without its contact to the matrix and nail bed. Thus, a triple combination of nail avulsion, topical therapy of the distal matrix and nail bed as well as systemic antifungal treatment was necessary [27] to achieve a cure in thirty-four digits and improvement in another three nails. Topical treatment was performed with either ciclopirox in a common lacquer base or a chitosan nail varnish as well as with amorolfine lacquer that was applied once weekly. The systemic drugs used were mainly terbinafine 250 mg daily and itraconazole 200 mg daily continuously or 400 mg daily for one week once a month [28]. Although the single-digit onychomycosis was difficult to treat, drug resistance was not the problem [29]. Newer systemic antifungals were not available [30]. Chemical avulsion using 40% urea paste is a very effective method [31]; however, it was not successful in most of our cases, perhaps because this method does not sufficiently remove the infected nail from under the proximal nail fold. Systemic treatment was first considered excessive by several referring dermatologists as they thought a single nail would not need such a long-time expensive and not always harmless therapy.

It is a well-known observation that treatment of onychomycosis may result in mycological and clinical cure in all nails but one [26]. The reason for the recalcitrant nature of the fungal infection usually remains obscure. It is tempting to speculate that a similar mechanism might apply. Furthermore, some nail tumors that commonly involve only one digit are relatively frequently infected with fungi, particularly onychomatricoma and onychopapilloma [32].

## 5. Conclusions

Recalcitrant onychomycosis of a single digit, mostly a fingernail, was shown to pose considerable diagnostic and therapeutic challenges. Common mycologic examinations remained mostly negative and generally accepted treatments for a single nail remained unsuccessful. It is important to ask for a previous trauma, confirm the diagnosis by established mycologic methods plus histopathology and institute an effective intensive long-term antifungal treatment to cure the nails.

## 6. Take-Home Message

▪Alterations of a single nail raise the suspicion of either a tumor or a trauma. The latter is usually remembered by the patient as this is commonly a painful event;▪Involvement by a fungal infection of a single nail may have been triggered by a trauma that has long been forgotten;▪The patient should be specifically asked for a previous trauma and mycologic examinations should be performed, as they may remain negative, the routine direct microscopy and culture should be complemented with histopathology or, if available, fungal PCR;▪The pre-damaged nail may pose a considerable therapeutic challenge.

## Figures and Tables

**Figure 1 jof-09-00313-f001:**
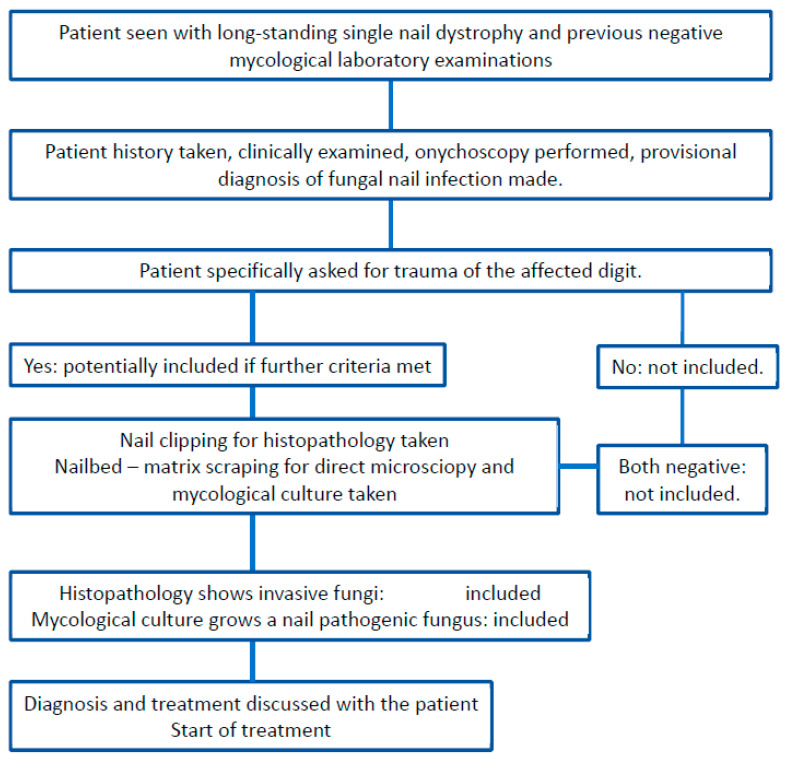
Flow chart of the patient inclusion process.

**Figure 2 jof-09-00313-f002:**
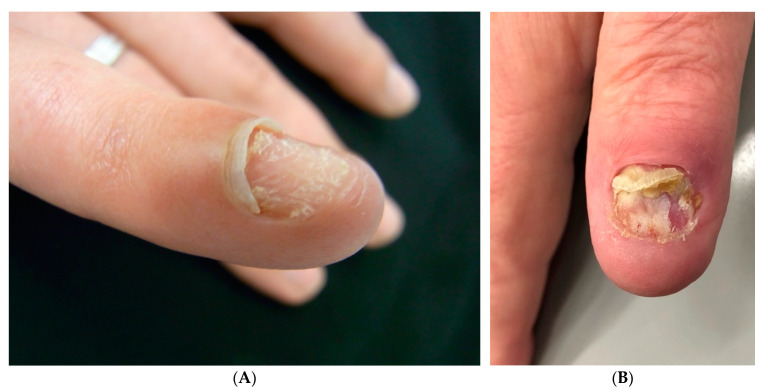
Single-digit onychomycosis. (**A**) Index finger of a 35-year-old woman. (**B**) Thumb of a 65-year-old man.

**Figure 3 jof-09-00313-f003:**
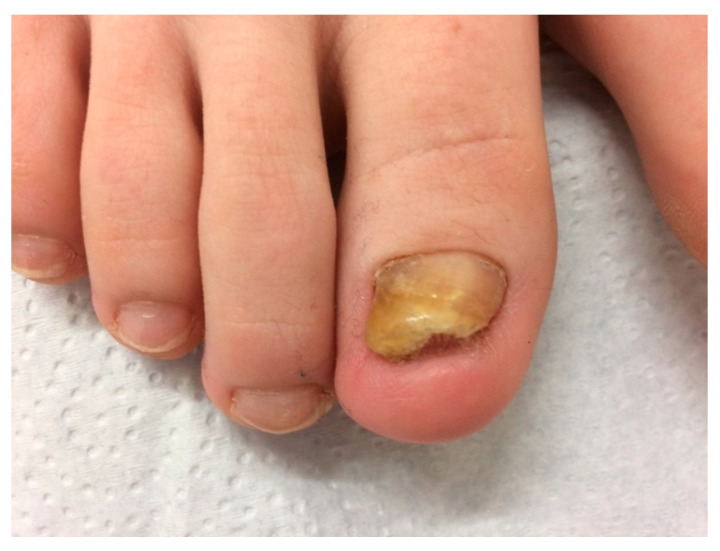
Post-traumatic single-digit onychomycosis of the big toe in a 28-year-old woman.

**Figure 4 jof-09-00313-f004:**
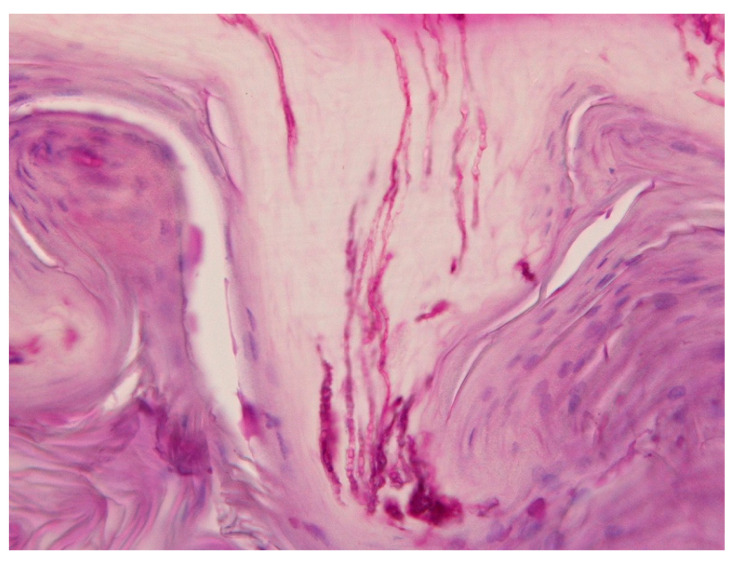
Histopathology of a nail clipping from the index finger of a 39-year-old female patient showing long slender filaments suggestive of dermatophyte hyphae. PAS stain, original magnification 400×.

**Figure 5 jof-09-00313-f005:**
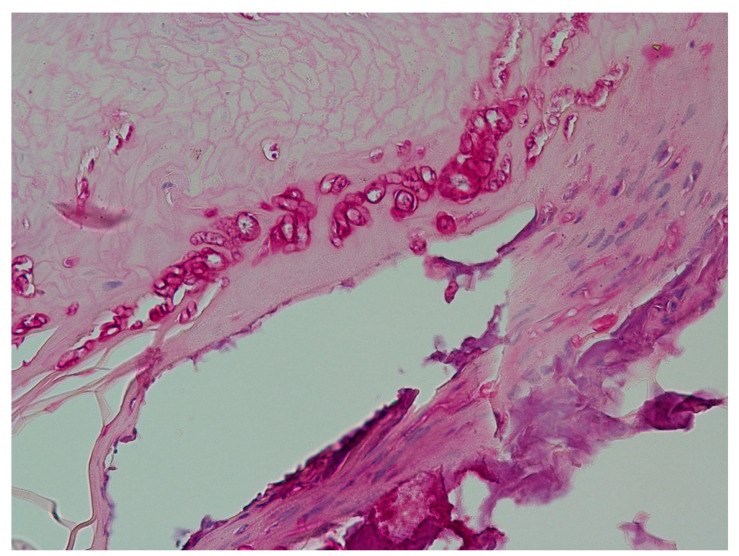
Histopathology of the nail clipping of a single-digit onychomycosis in a 43-year-old Tamil female suggestive of a non-dermatophyte onychomycosis. PAS stain, original magnification 400×.

**Figure 6 jof-09-00313-f006:**
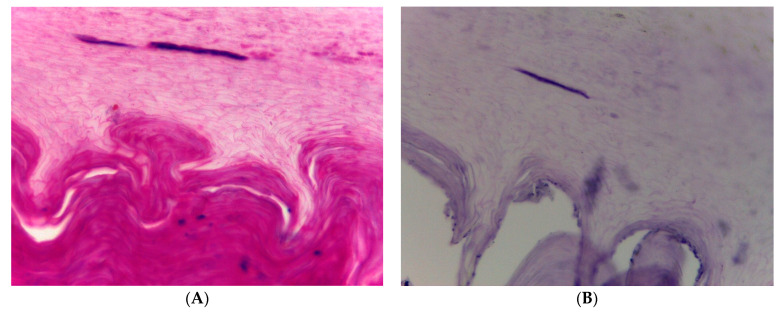
Nail clipping histopathology from the middle finger of a 25-year-old woman with single-digit onychomycosis. (**A**) Hematoxylin and eosin stain shows a basophilic fungal filament in the nail plate, (**B**) demonstrates the same area in PAS stain. Culture had grown Fusarium solani. Original magnification 200×.

**Figure 7 jof-09-00313-f007:**
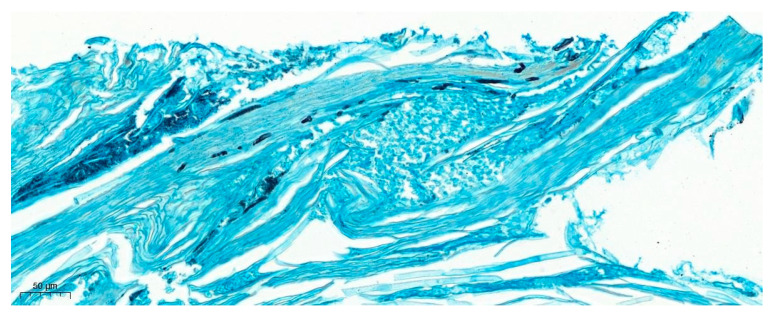
Histopathology of the nail bed punch biopsy of a 33-year-old female patient with single-digit post-traumatic onychomycosis. Grocott methene amine silver stain demonstrates slender hyphae suggestive of a dermatophyte infection, 200×.

**Figure 8 jof-09-00313-f008:**
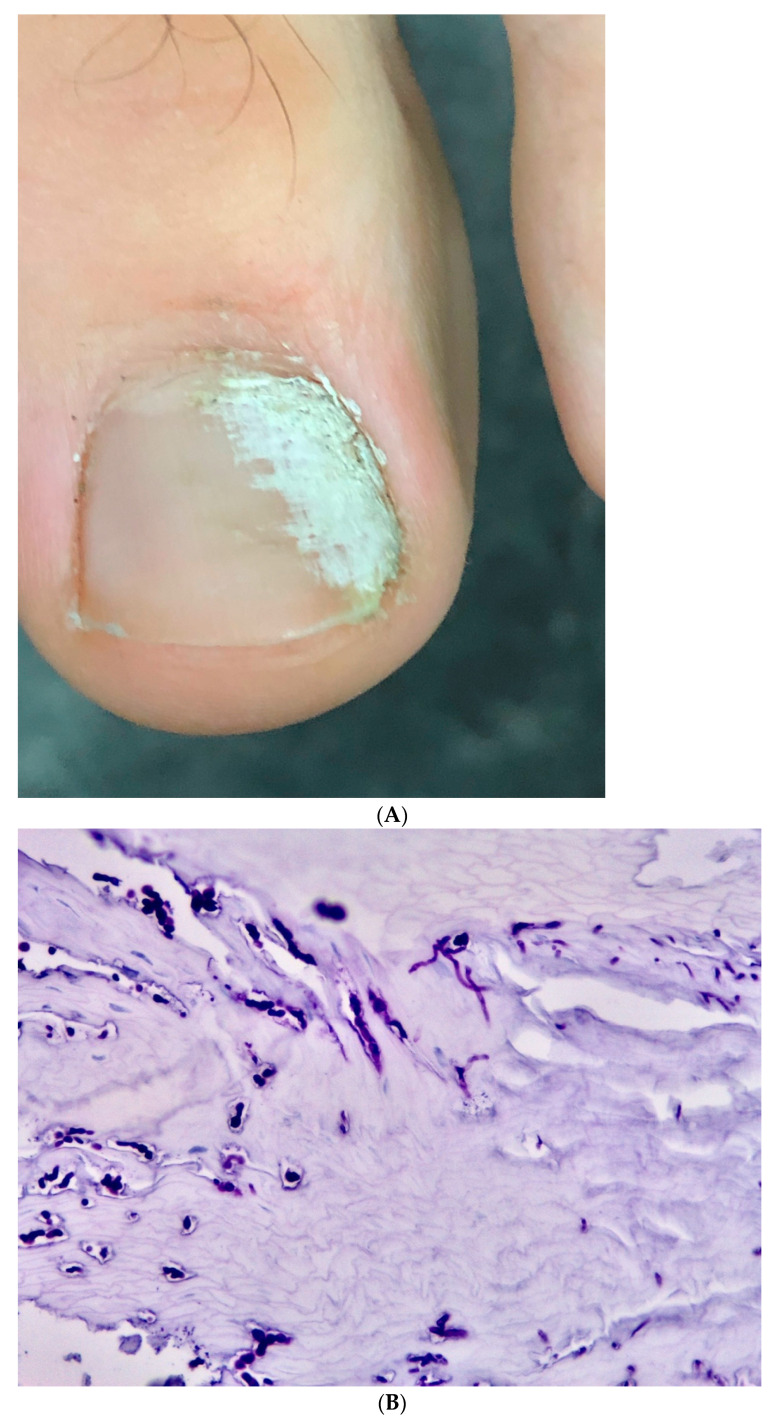
Post-traumatic single-digit onychomycosis of the big toe due to *Fusarium solani*. (**A**) Clinical picture, (**B**) histopathology of the nail clipping showing fungal filaments of variable diameter and spore-like elements; PAS, original magnification 200×.

**Table 1 jof-09-00313-t001:** Demographic data.

		*p* Value
Gender	F 30, M 14, total 44	ns (0.9987)
Age range	22–68 years	
Affected nail	Finger 39, toe 5	*p* = 0.00116
Previous mycologic cultures	Negative 40 (90.9%) Trichophyton rubrum 2 (4.5%) Non-dermatophyte mold 2 (4.5%)	*p* < 0.001
Previous antifungal treatment	34 with no substantial improvement (77.2%)	
Current diagnosis confirmed	44 by histopathology (100%) 30 by culture (68.2%): 14 *T. rubrum* (31.8%) 1 *T. mentagrophytes* complex (2.3%) 15 non-dermatophyte molds (34.1%)	*p* < 0.0001 *p* = 0.006363

**Table 2 jof-09-00313-t002:** Advantages of histopathology for the diagnosis of onychomycoses.

High sensitivity—particularly in difficult to diagnose cases.
Lack of contamination.
Permanent preparation.
Simple procedure.
Demonstration of fungal invasion of the nail organ.
Identification of other or concomitant nail disorders.
Considerably quicker results than fungal culture.

**Table 3 jof-09-00313-t003:** Techniques for the identification of fungal nail infections [16,17,18,19,20,21].

Method	Advantage	Disadvantage
KOH without or with Parker ink or blancophore	Easy, rapid, cheap.	Relatively non specific, requires expertise. Blancophore requires fluorescence microscope.
Culture	Identification of fungus.	No differentiation between pathogen and colonizer. High percentage of biologically false negatives.
Histopathology (PAS, Grocott)	Differentiation of type of fungal infection, high yield of positives, differentiation between invasive onychomycosis and colonization, differential diagnoses possible.	No species identification.
Immunohistochemistry	Fungus identification?	Difficult on nail sections, few antibodies available.
Polymerase chain reaction	Identification of fungus, rapid.	No differentiation between pathogen and colonizer, expensive.
Immunochromatography	Easy to perform. Useful supplement for KOH.	Only for dermatophytes (*T rubrum*).
In situ hybridization	Localization of infection and pathogen identification (?).	No routine primers available.
MALDI-ToF mass spectroscopy	Species identification possible after culture.	Requires previous culture. Depends on fungal protein library.
Optical coherence tomography	Fungi may be detected.	Expensive. No species identification.
Confocal laser scanning microscopy	Fungi can be identified.	Expensive. No species identification.
Ultraviolet fluorescence excitation imaging	Immediate diagnosis of fungus infected nails possible.	Not yet available.

## Data Availability

Not applicable.

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
