# Peer review of "Post-Traumatic Single-Digit Onychomycosis"

_jof, 2023, doi:10.3390/jof9030313_

Round 1
Reviewer 1 Report
Very good paper
Author Response
Thank you very much for your kind comment.
Reviewer 2 Report
The topic is one of importance given the high number of presentations to health services that are related to concerns on
the prevalence and related factors of post-traumatic single-digit onychomycosis. Also, this is an a group of fungal nail infections commonly classified either according to the pathogenic fungus, to the duration of the disease or to the mode of fungal invasion. . I think it would be a more clear study if the following parts were revised and supplemented. These will be discussed below relative to the information of the manuscript.
General Comments:
Overall the manuscript is generally well written and is a topic of interest. However after reading it a number of times I am still left without key take-home points. I believe these points are in the results they just need to be developed.
Specific comments:
Abstract:
1) The authors state most cases are diagnosed clinically although there is a general consensus that the pathogen should be identified prior to initiating a treatment. This seems like too much of an over simplification of what was done. I do feel that it would be beneficial to explain what specifically you are looking at in relation to objective of this research related with (this also applies to the main body of the paper). Is it the development of post-traumatic onychomycosis literature. This needs to be made clearer throughout the paper. (Major Compulsory Revision)
Introduction
2) The first paragraph should have a sentence or two added that better outlines why this study is important related with post-traumatic onychomycosis
Furthemore, the authors do a poor job on reviewing relevant literatura related with importance with onychomycosis and quality of life. Please revise the research of Evaluation of foot health related quality of life in individuals with foot problems by gender: a cross-sectional comparative analysis study https://pubmed.ncbi.nlm.nih.gov/30341140/
3) In the last paragraph, the significance of the proposed word should be included highlighting why your work is important. what is the scientific contribution of this paper? it is not clear how this paper can make a significant contribution to the state of the art. (Major Compulsory Revision).
In addition, author´s hypotheses should be included.
4) This methods section is poor, needs to present a better rationale for the study and the methodology employed. Also, neither appear information related with inclusion and exclusion criteria, dates, protocol. The study design is a experimental research of ramdom sampling method, where the study was conducted in the hospital or in the university center? This research adhere to reporting CONSORT guidelines? (Major Compulsory Revision).
6) Where the experiments carried out? In a hospital? In an educational institution? Provide this information.
7) Add figure 1 as a study flow chart for the readers. (Major Compulsory Revision).
8) Include p-values in all the tables (Major Compulsory Revision).
9) The Discussion section is a rehashing of the results. It does not appear that the authors include much interpretation of what the study findings mean for clinical practice or research. Please remove table 2 and 3 this info not is relevant in this section (Major Compulsory Revision)
FInally, the conclusión is weak and too long. (Major Compulsory Revision)
Author Response
Thank you for your comments.
I have made the changes requested, please see below.
“Specific comments:
Abstract:
1) The authors state most cases are diagnosed clinically although there is a general consensus that the pathogen should be identified prior to initiating a treatment. This seems like too much of an over simplification of what was done. I do feel that it would be beneficial to explain what specifically you are looking at in relation to objective of this research related with (this also applies to the main body of the paper). Is it the development of post-traumatic onychomycosis literature. This needs to be made clearer throughout the paper. (Major Compulsory Revision)”
Authors‘ response
As stated in the abstract and in the text this report describes a subset of patients with long-term undiagnosed single digit nail disease that turned out to be onychomycosis. I fully agree with the general consensus that the pathogen should be identified prior to initiating treatment; however, the problem in these patients was that the process of pathogen identification had failed in almost all cases as both direct microscopy as well as culture had been (repeatedly in most cases) negative. It was thus the senior author’s clinical suspicion and perseverance to insist on further efforts to find the pathogen. The essence of this point is to take a detailed history and ask for a previous trauma as well as to intensify the laboratory work-up to prove the fungal nature of the infection. We have shown that we do not simply diagnose and treat on clinical grounds alone. (This contradicts the reviewer’s assumption that the authors simply base their diagnosis on clinical grounds and then treat right away.)
‘Most cases are diagnosed clinically although there is a general consensus that the pathogen should be identified prior to initiating a treatment’ has been re-phrased to ‘The diagnosis of onychomycoses should be proven by laboratory methods prior to initiating a treatment although in general practice, they are often treated if this fails.’
Although trauma has been accepted as an important predisposing factor for onychomycoses a Pubmed search for posttraumatic (post-traumatic) onychomycosis did not yield a single hit related to fungal nail infection (the only hit was on traumatic septic arthritis). Thus, this appears to be a rarely considered though clinically important event in the development of onychomycoses. As suggested, this is now included into the abstract.
The importance of diagnosing a single digit posttraumatic onychomycosis is now included into the abstract: ‘A single digit nail disease should prompt the clinician to ask for a previous trauma to this digit and to intensify the search for a specific pathogen.’
“Introduction
2) The first paragraph should have a sentence or two added that better outlines why this study is important related with post-traumatic onychomycosis”
Response
We have added two sentences to the introduction highlighting the clinical importance of posttraumatic single-digit onychomycosis and what should result from its diagnosis:
‘Although trauma in general is a well-known predisposing factor for the development of fungal nail infections a literature search in Pubmed did not yield any articles on onychomycosis after trauma. Alterations of a single nail, particularly of a fingernail in a female patient, should prompt the clinician to specifically ask for a trauma to this digit and to add histopathology of the nail or another sensitive method for the laboratory diagnosis of a fungal nail infection.’
“Furthemore, the authors do a poor job on reviewing relevant literature related with importance with onychomycosis and quality of life. Please revise the research of Evaluation of foot health related quality of life in individuals with foot problems by gender: a cross-sectional comparative analysis study https://pubmed.ncbi.nlm.nih.gov/30341140/”
The publication by López-López et al found that “women with foot problems present a negative impact on QoL related to foot health as compared with men except in the domains of Overall Health and Social Capacity, which appears to be associated with the presence of foot conditions.” However, the single-toe onychomycoses were all in men except one and this is the reason why this aspect was not specifically discussed in the manuscript.
López-López D, Becerro-de-Bengoa-Vallejo R, Losa-Iglesias ME, Palomo-López P, Rodríguez-Sanz D, Brandariz-Pereira JM, Calvo-Lobo C. Evaluation of foot health related quality of life in individuals with foot problems by gender: a cross-sectional comparative analysis study. BMJ Open. 2018 Oct 18;8(10):e023980
Onychomycoses have a severe impact on the quality of life. This has been researched for since more than 30 years. Fingernail involvement in women has a particularly serious impact on QoL. There are many publications discussing this aspect (Pubmed yields a 163 hits when searching for onychomycosis and quality of life). It was not the authors’ intention to discuss this undoubtedly important issue as this would go far beyond the scope of the topic of posttraumatic onychomycosis.
Drake LA, Patrick DL, Fleckman P, André J, Baran R, Haneke E, Sapède C, Tosti A. The impact of onychomycosis on quality of life: development of an international onychomycosis-specific questionnaire to measure patient quality of life. J Am Acad Dermatol 1999;41(2 Pt 1):189-196
Stewart CR, Algu L, Kamran R, Leveille CF, Abid K, Rae C, Lipner SR. Effect of onychomycosis and treatment on patient-reported quality-of-life outcomes: A systematic review. J Am Acad Dermatol 2021;85(5):1227-1239
However, the quality of life aspect is now included and the phrase changed to ‘Onychomycoses are not only aesthetically embarrassing and have a serious impact on the quality of life [5] but may pose a true medial risk, particularly in diabetics, patients under immunosuppression and cancer therapy [6]. Further, they may be the portal of entry for streptococcal cellulitis and the origin of recurrences for tinea pedum.
Ref 5 is now Stewart CR, Algu L, Kamran R, Leveille CF, Abid K, Rae C, Lipner SR. Effect of onychomycosis and treatment on patient-reported quality-of-life outcomes: A systematic review. J Am Acad Dermatol 2021;85(5):1227-1239
“3) In the last paragraph, the significance of the proposed word should be included highlighting why your work is important. what is the scientific contribution of this paper? it is not clear how this paper can make a significant contribution to the state of the art. (Major Compulsory Revision).”
Response: The role of the trauma to a single digit has to the best of our knowledge not systematically been investigated. This small study underlines its importance. Larger studies are warranted.
“In addition, author´s hypotheses should be included.”
Response: The exact mechanism by which a trauma that might have been sustained years ago, renders this nail particularly susceptible for fungal infection is not known. It might by hypothesized that posttraumatic neuro-vascular alterations may impair the blood supply and reduce the immune response to fungal infections in the affected region. In extreme cases, a simple trauma like a nail biopsy may even lead to reflex sympathetic dystrophy, a variant of the complex regional pain syndrome [7].
Qureshi AA, Friedman AJ. Complex regional pain syndrome: What the dermatologist should know. J Drugs Dermatol 2018;17(5):532-536
“4) This methods section is poor, needs to present a better rationale for the study and the methodology employed. Also, neither appear information related with inclusion and exclusion criteria, dates, protocol. The study design is a experimental research of ramdom sampling method, where the study was conducted in the hospital or in the university center? This research adhere to reporting CONSORT guidelines? (Major Compulsory Revision).“
Response to the reviewer:
The study is based on the senior author’s clinical experience. The first cases date back 15 years. At this time it was not clear that these observations might be more than a few chance observations in the author’s practice. Thus it is not a prospective study with inclusion and exclusion criteria etc.
As mentioned in this section, the cases were seen over a period of 15 years, thus this is a random sampling method. There were no other exclusion criteria than lack of proof of fungal infection and lack of a trauma to this specific digit in the patients’ histories. The patients were seen during specialized nail clinics in the Department of Dermatology, Inselspital, University of Berne, Switzerland, in a private dermatology clinic in Freiburg, Germany, in the Department of Dermatology, University Hospital Ghent, Gent, Belgium, in a large practice clinic of Centro de Dermatología Epidermis, Instituto CUF, Porto, Portugal, and 2 were seen in a large Dermatology Center in Kiyv, Ukraine (by AS).
There was no particular protocol beyond good clinical practice to diagnose a fungal nail infection: patient history, question for risk factors, clinical examination including onychoscopy, direct microscopy, mycological culture and histopathology of nail specimens with PAS and Grocott in 2 cases. If this is relevant for the manuscript I will be happy to include this.
As this study represents a series of observations over a period of 15 years and started before CONSORT guidelines even existed and is not a randomized clinical trial (“CONSORT Statement … is an evidence-based, minimum set of recommendations for reporting randomized trials”) the CONSORT guidelines do not apply.
I have added: ‘ The inclusion criteria were positive history of trauma to the affected digit and proof of fungal infection by mycological culture and/or histopathology. There were no specific exclusion criteria.’
“6) Where the experiments carried out? In a hospital? In an educational institution? Provide this information.“
Response: The patients were seen during specialized nail clinics in the Department of Dermatology, Inselspital, University of Berne, Switzerland, in a private dermatology clinic in Freiburg, Germany, in the Department of Dermatology, University Hospital Ghent, Gent, Belgium, in a large practice clinic of Centro de Dermatología Epidermis, Instituto CUF, Porto, Portugal, and 2 were seen in a large Dermatology Center in Kiyv, Ukraine (by GS). This is indicated in the affiliations of the authors.
7) Add figure 1 as a study flow chart for the readers. (Major Compulsory Revision).
Response: A flow chart is added as figure 1 (see below).
8) Include p-values in all the tables (Major Compulsory Revision).
Response: As the numbers were self evident and not large we had originally not included p-values. P values are now included in the table where feasible.
9) The Discussion section is a rehashing of the results. It does not appear that the authors include much interpretation of what the study findings mean for clinical practice or research. Please remove table 2 and 3 this info not is relevant in this section (Major Compulsory Revision)
Response:
The discussion is shortened and the results are discussed where they are needed for the justification or explanation.
Tables 2 and 3 are removed.
Paragraph does not contain what has been described in the results section.
“Finally, the conclusión is weak and too long. (Major Compulsory Revision)“
The conclusion summarizes that single digit nail changes may be traumatically induced onychomycosis and should prompt the clinician to perform mycologic tests before embarking on a systemic treatment.
Reviewer 3 Report
The authors describe 44 cases of onychomycosis with different periods of evolution that were referred to the specialty center after several unsuccessful attempts at treatment in other medical centers.
The cases are well documented and the results are well showed and discussed. Tables are well organized and the figures are illustrative.
The subject is interesting not only for specialists, but also for general practitioners and dermatologists who frequently see these patients. and in this context, some information and discussions can be useful for the patients’ case management.
As a suggestion, it would be interesting to have more information about the cases where no complete healing was observed. Were there any associated comorbidities? Any uncontrolled habit (shoes, constant trauma, etc.) that would facilitate the maintenance of the infection? What was the follow-up given to these patients? New treatments?
general comments:
1- table 1- include even if in the footer of the table the indication of which non-dermatophytes were identified (in the text it informs that the majority were of the genus Fusarium but general information would be interesting);
2- as regard these cases of diagnosis of infection by Fusarium spp. Did they have any particularity that facilitated the presence of this fungus?
3- MALDI-ToF- nowadays the cost of this technique has dropped, but the access to specific protein libraries is still a major limitation. Maybe it could be discussed in the text and presented in table 3
4- All histopathology figures (3-7) should have a bar and arrows, thus facilitating identification by non-specialists. in addition, they must have in the legends the identification of the stain color used and the magnitude. Some has partially this information but they should be included in all of them.
5- Normally, the use of patient data requires ethical clearance, but the authors do not mention this information, or explain the lack of such document, if applicable.
Author Response
Thank you for comments and suggestions. Please find my answers below.
As a suggestion, it would be interesting to have more information about the cases where no complete healing was observed. Were there any associated comorbidities? Any uncontrolled habit (shoes, constant trauma, etc.) that would facilitate the maintenance of the infection? What was the follow-up given to these patients? New treatments?
Thank you for your comments.
The cases without complete cure were not different from the other ones, there were no specific comorbidities rendering them more susceptible to infection (no particular immunosuppression, vascular impairment, neural disease, etc). All patients were instructed to disinfect their shoes on a regular basis, change their socks daily, avoid ill-fitting shoes, etc. Further, it is difficult to follow-up patients over years for a nail infection and compliance is always a problematic issue. Further, a 100% clinical plus mycological cure is not achieved in any study as there are too many factors influencing complete cure. As the senior author has been practicing in different countries for variable periods of time the follow-up periods were between 3 and a maximum of 12 years. As stated all patients received a combined therapy of nail avulsion, systemic terbinafine for dermatophytosis and itraconazole for non-dermatophyte onychomycosis. No experimental drugs were applied.
general comments:
- table 1- include even if in the footer of the table the indication of which non-dermatophytes were identified (in the text it informs that the majority were of the genus Fusarium but general information would be interesting);
The laboratories changed over time. The early cases were culture proven and dermatophytes were determined to the species level as indicated. The mycology lab (Univ Bern) changed from cultures to PCR in 2015-2016 and there was a period with incertainties when Fusarium spp were diagnosed before they were determined more precisely to F solani and oxysporum complex. Common molds were not accepted as nail pathogens.
- as regard these cases of diagnosis of infection by Fusarium Did they have any particularity that facilitated the presence of this fungus?
Much to our surprise, there were no special clinical particularities in these cases. Fusarium was thought first to be very uncommon (if not impossible) among our patients; however, there is a recent publication on Fusarium onychomycoses in Switzerland, and we have now a high percentage of Fusarium OM (see ref 15).
- MALDI-ToF- nowadays the cost of this technique has dropped, but the access to specific protein libraries is still a major limitation. Maybe it could be discussed in the text and presented in table 3
I have omitted table 3 at the request of reviewer 2. However, MALDI-ToF is a rapid method once the column is available, and the labs having this technology apparently like it (the lab in Freiburg, Germany, uses MALDI-ToF). This is briefly mentioned in the discussion.
- All histopathology figures (3-7) should have a bar and arrows, thus facilitating identification by non-specialists. in addition, they must have in the legends the identification of the stain color used and the magnitude. Some has partially this information but they should be included in all of them.
The stains and magnifications are now included.
- Normally, the use of patient data requires ethical clearance, but the authors do not mention this information, or explain the lack of such document, if applicable.
Most patients were seen at a time when there was no such ethical clearance was required. All newer patients gave their verbal permission as this is part of the patient admission procedure.
Round 2
Reviewer 2 Report
The authors have clearly and adequately addressed all comments raised by the reviewers. Please also consider adding the information that appear as "Take home message" as text in the discussion section.
Author Response
Thank you for your thoughtful criticisms.
Our answers have addressed your points, However, according to reviewers 1 and 3, I have re-added tables 2 and 3 as this appeared interesting for them.
I added p values plus percentages in table 1 where adequate.
The importance of the observations has been stressed in the discussion. I have added a line concerning the role of trauma in the surgical treatment of onychomycosis by nail avulsion.
The QoL issue has not been extended as there are many systematic studies on this topic; however, the repeated consultations of the patients with different dermatologists underlines that they experienced a decrease in life quality.